# Human Ocular Toxoplasmosis in Romania: History, Epidemiology, and Public Health: A Narrative Review

**DOI:** 10.3390/microorganisms12081541

**Published:** 2024-07-27

**Authors:** Laura Andreea Ghenciu, Ovidiu Alin Hațegan, Sorin Lucian Bolintineanu, Alexandra-Ioana Dănilă, Roxana Iacob, Emil Robert Stoicescu, Maria Alina Lupu, Tudor Rareș Olariu

**Affiliations:** 1Department of Functional Sciences, “Victor Babes” University of Medicine and Pharmacy Timisoara, Eftimie Murgu Square No. 2, 300041 Timisoara, Romania; bolintineanu.laura@umft.ro; 2Center for Translational Research and Systems Medicine, “Victor Babes” University of Medicine and Pharmacy Timisoara, Eftimie Murgu Square No. 2, 300041 Timisoara, Romania; 3Discipline of Anatomy and Embriology, Medicine Faculty, “Vasile Goldis” Western University of Arad, Revolution Boulevard 94, 310025 Arad, Romania; 4Department of Anatomy and Embriology, “Victor Babes” University of Medicine and Pharmacy Timisoara, 300041 Timisoara, Romania; s.bolintineanu@umft.ro (S.L.B.); alexandra.danila@umft.ro (A.-I.D.); roxana.iacob@umft.ro (R.I.); 5Doctoral School, “Victor Babes” University of Medicine and Pharmacy Timisoara, Eftimie Murgu Square No. 2, 300041 Timisoara, Romania; 6Field of Applied Engineering Sciences, Specialization Statistical Methods and Techniques in Health and Clinical Research, Faculty of Mechanics, ‘Politehnica’ University Timisoara, Mihai Viteazul Boulevard No. 1, 300222 Timisoara, Romania; stoicescu.emil@umft.ro; 7Department of Radiology and Medical Imaging, “Victor Babes” University of Medicine and Pharmacy Timisoara, Eftimie Murgu Square No. 2, 300041 Timisoara, Romania; 8Research Center for Pharmaco-Toxicological Evaluations, “Victor Babes” University of Medicine and Pharmacy Timisoara, Eftimie Murgu Square No. 2, 300041 Timisoara, Romania; 9Discipline of Parasitology, Department of Infectious Disease, “Victor Babes” University of Medicine and Pharmacy Timisoara, 300041 Timisoara, Romania; lupu.alina@umft.ro (M.A.L.); rolariu@umft.ro (T.R.O.); 10Center for Diagnosis and Study of Parasitic Diseases, “Victor Babes” University of Medicine and Pharmacy Timisoara, 300041 Timisoara, Romania

**Keywords:** ocular toxoplasmosis, *Toxoplasma gondii*, seroprevalence, uveitic patients

## Abstract

Toxoplasmosis, caused by the protozoan parasite *Toxoplasma gondii* (*T. gondii*), presents a significant global health concern, particularly for immunocompromised individuals and congenitally infected newborns. Despite its widespread prevalence, there are limited data on *T. gondii* seroprevalence and ocular toxoplasmosis in Romania. This review aims to summarize the research accomplished on the prevalence and epidemiology of human ocular toxoplasmosis in Romania. Ocular toxoplasmosis, a leading cause of infectious posterior uveitis worldwide, involves complex interactions between host immune responses and parasite factors. Clinically, it presents as focal necrotizing retinitis, characterized by active focal retinal lesions with adjacent chorioretinal scarring, often accompanied by vitreous inflammation and anterior chamber reactions. Diagnosis relies on clinical examination supported by fundus photography, optical coherence tomography (OCT), and serological assays. The authors followed the Preferred Reporting Items for Systematic Reviews and Meta-Analyses (PRISMA) standards, conducting a literature review on PubMed, Google Scholar, and Scopus. Our focus was on ocular toxoplasmosis in Romania, and we used keywords and specific MeSH terms. Finally, 17 articles met all the criteria, as summarized in the PRISMA diagram. This study underscores the need for improved diagnostic methods, increased research efforts, and comprehensive public health education to mitigate the burden of toxoplasmosis and ocular toxoplasmosis in Romania.

## 1. Introduction

Toxoplasmosis, caused by the obligate intracellular protozoan parasite *Toxoplasma gondii* (*T. gondii*), represents a significant global health concern [1]. While the majority of infections in immunocompetent individuals are asymptomatic or present as mild flu-like symptoms, toxoplasmosis poses considerable risks, particularly in immunocompromised individuals and congenitally infected newborns [2]. 

T. gondii, first discovered in 1908 by Nicolle and Manceaux in a North African rodent, gained recognition as a significant human pathogen only in the mid-20th century following the development of the Sabin–Feldman dye test. Epidemiologically, *T. gondii* is globally distributed, with varying prevalence rates depending on geographic location, socioeconomic factors, and cultural practices. Seroprevalence studies indicate that a significant portion of the world’s population has been exposed to the parasite. In regions with poor sanitation and hygiene, prevalence rates tend to be higher due to increased exposure to oocysts exposed by infected cats, the primary definitive hosts of *T. gondii* [1,2,3].

There is little information regarding the seroprevalence of *T. gondii* in Romania and even fewer studies on ocular toxoplasmosis. As far as is known, Romania does not have any statistically reliable studies on the epidemiology of ocular toxoplasmosis. The majority of epidemiological data were gathered retroactively, without statistical significance being determined. Most research showed that living in a rural area and age increased the overall seroprevalence of *T. gondii*. The majority of analyses relied on convenience samples; however, Coroiu et al. [4] conducted research on an important number of samples from the general population of 11 counties in Romania. *T. gondii* IgG antibodies were detected in these samples using two standard tests (Enzyme-linked immunosorbent13-ELISA13, Latex agglutination test2-LAT2), with comparable outcomes. From the samples tested, seropositivity was 59.4%, and, depending on the county, patients had *T. gondii* antibodies within a range of 44.9% to 70.2%. There are further studies that assessed *T. gondii* infection in humans for different purposes. Olariu et al. [5,6] demonstrated a prevalence of 64.8% in the adult population and 55.8% in pregnant women from Western Romania, while Lupu et al. [7] demonstrated a *T. gondii* seroprevalence of 45.9% in Romanian blood donors, with a significant age-associated increase.

While specific data on *T. gondii* prevalence in Romania may vary, studies suggest that the parasite is endemic in the country, with seroprevalence rates among the population likely reflecting those observed in other European countries. In Romania, as in many regions worldwide, factors such as climate, socioeconomic conditions, cultural practices, and population density influence the epidemiology of *T. gondii*. Practices related to agriculture, particularly the consumption of undercooked meat from infected animals, and exposure to contaminated soil or water can contribute to human infection. Additionally, the presence of domestic cats, the primary hosts of *T. gondii*, contributes to environmental contamination with oocysts, further increasing the risk of human exposure [1,2,3]. Results of a recent study conducted in 1347 blood donors revealed that age, level of education, and having pets (cats and/or dogs) were factors significantly associated with *T. gondii* in Romania [7].

Globally, the prevalence of *T. gondii* infection and ocular toxoplasmosis varies; the estimated combined prevalence of ocular toxoplasmosis in the general population is 2%. The United States of America had the highest occurrence rate, 6%, and the reported prevalence rate in patients with uveitis was 9%. Prevalence rates were greater in middle-income nations. Compared to panuveitis, which had a prevalence of 7%, posterior uveitis had a much greater prevalence of ocular toxoplasmosis [8]. There are regions of the world, notably southern Brazil, where the prevalence of ocular involvement is significantly higher [9]. According to earlier research, the incidence of ocular toxoplasmosis in Latin America ranges from three cases per 100,000 people in Colombia to 26.2 active cases per 100,000 people in Cuba [10,11]. According to German investigators, there are 1.5–2.5 incidences for every 100,000 people nationwide [12]. The variability in approach makes it impossible to compare the research, although it is clear that there is a variation in the number of new cases.

Ocular toxoplasmosis represents a leading cause of infectious posterior uveitis worldwide. The pathogenesis of ocular toxoplasmosis is multifactorial, involving a complex interplay between host immune responses and parasite factors [3]. Following reactivation of latent infection or primary acquisition, *T. gondii* invades the retina, inducing a localized inflammatory response. This response is characterized by the recruitment and activation of immune cells, cytokine release, and tissue destruction. The parasite’s ability to evade host immune defenses, modulate host signaling pathways, and establish chronic infection contributes to the chronicity and recurrences observed in ocular toxoplasmosis [13].

Clinically, ocular toxoplasmosis presents as focal necrotizing retinitis, typically characterized by active focal retinal lesions with adjacent chorioretinal scarring. Lesions vary in size, location, and morphology and may be associated with vitreous inflammation and anterior chamber reactions [14]. Diagnosis relies on clinical examination supported by ancillary tests such as fundus photography, optical coherence tomography (OCT), and serological assays to detect specific anti-Toxoplasma antibodies [15].

Management of ocular toxoplasmosis aims to control active inflammation, prevent disease progression, and minimize loss of visual acuity. Treatment strategies include anti-parasitic agents, such as pyrimethamine and sulfadiazine, to target active infection, and these are often supplemented with adjunctive corticosteroids to reduce inflammation. However, therapeutic decisions are nuanced, considering factors such as lesion characteristics, location, and associated complications [16].

Despite therapeutic interventions, ocular toxoplasmosis may lead to irreversible visual impairment, including macular scarring, retinal detachment, and optic nerve atrophy. Moreover, the risk of recurrence and chronicity underscores the need for long-term follow-up and proactive management strategies. Prevention strategies, including public health education, avoidance of raw or undercooked meat, and proper hygiene practices, remain crucial in mitigating the burden of ocular toxoplasmosis [17].

Therefore, this study specifically reviewed the prevalence and epidemiology of human ocular toxoplasmosis in Romania, providing a comprehensive overview of the disease, by incorporating the available publications regarding this topic. The primary objective was to highlight the significance of human ocular toxoplasmosis, which remains notably understudied and underdiagnosed in Romania. By synthesizing current data and identifying gaps in research and diagnosis, this review emphasizes the need for improved diagnostic methods, increased research efforts, and comprehensive education on human ocular toxoplasmosis. This knowledge could also be useful to healthcare professionals and researchers, encouraging further studies and public health initiatives focused on this overlooked condition.

## 2. Materials and Methods

The Preferred Reporting Items for Systematic Reviews and Meta-Analyses (PRISMA) standards were followed during the selection of studies that were included in this research [18]. The current literature review is based on bibliographic searches conducted using MeSH terms (on PubMed) and both manual and automated searches in the PubMed database, Google Scholar, and Scopus. Based on their title, the abstract, and a brief look at the entire research manuscript, the most relevant articles were selected. All of the articles selected were added to a Microsoft Excel table with the following columns for improved subsequent management and organization of the review: title, authors, year and journal of publication, type of publication (case reports were also included), keywords, and type of ocular affection. Publications with too little an amount of relevant information, duplicates, and articles published in languages other than English were also eliminated. Most relevant publications were chosen based on the fact that our main aim was the study of ocular toxoplasmosis in Romania.

Initially, the research studies were hand-searched in the above-mentioned databases using the following keywords: “ocular toxoplasmosis in Romania”, “retinochoroiditis toxoplasma in Romania”. Subsequently, a second search was carried out using the MeSH term option present in PubMed with the following terms: ((“Ocular”[Mesh]) AND “Toxoplasmosis”[Mesh]) AND “Romania”[Mesh]; ((“Eye”[Mesh]) AND “Toxoplasmosis”[Mesh]) AND “Romania”[Mesh]; ((“Retinochoroiditis”[Mesh]) AND “Toxoplasmosis”[Mesh]) AND “Romania”[Mesh].

Overall, a very small number of articles described cases of ocular toxoplasmosis, many of which were published decades ago. The scientific community in Romania has well described the prevalence of acquired toxoplasmosis in the last few years but has failed to describe one of the most important complications, ocular toxoplasmosis. Finally, 17 articles were chosen for the literature review since they fulfilled all the requirements. The process followed for selecting the articles for this review is summarized in the diagram below—a PRISMA diagram (Figure 1).

## 3. History and Epidemiology-State of the Art in Romania

In 1956, Dragomir et al. [19] published the first paper on *T. gondii* in Romania, highlighting a case of congenital human toxoplasmosis. Several other researchers have since published their findings regarding toxoplasmosis in humans and animals in Romania in the 20th century. Numerous reports have been made since then, most of which involve immunological studies of fertile women as part of their screening investigations [20]. 

There are multiple reports of clinical and serological testing performed on patients with suspected ocular toxoplasmosis in Romania [21,22,23]. Dogan and Farah [24] identified ocular toxoplasmosis in 21.9% of pediatric cases with uveitis. According to Teodorescu et al. [21], of the 135 hospitalized cases of toxoplasmosis (acquired or congenital) in different hospitals in Bucharest, 66.7% cases had ocular implication. According to Lazar et al. [25], ocular toxoplasmosis is generally thought to be frequent in the Romanian population, although data on the frequency of ocular diseases in the general public are lacking. There have also been several case-reports published in recent decades, many of which describe very interesting cases from Romania. These manuscripts describe atypical forms of ocular toxoplasmosis that feature little to no apparent retinal changes [26,27]. Stanila [28] has conducted very important research during which he examined 22 hospitalized patients within a two year time-frame whose clinical diagnoses had been verified by *T. gondii* seropositivity. He observed that the likelihood of starting a specific treatment depends on two factors: the antibodies (IgG and IgM) and the ocular symptoms. *T. gondii* chorioretinitis is an uncommon form of toxoplasmosis that is thought to exclusively occur in immunocompromised or severely infected adult patients. However, in recent years, it has been observed to occur more frequently in immunocompetent patients [29]. One study has described a group of 53 patients with retinouveal disease, out of which 84.9% resulted in active ocular toxoplasmosis cases and responded very well to the classic treatment scheme [25]. A diagnosis of ocular toxoplasmosis in Romania was usually made by assessing the positive serology results with the clinical data [27,30,31]. Table 1 describes reports on ocular toxoplasmosis in Romania from the last two decades; case studies or studies that did not contain all of the information mentioned below were not included in this table. 

Serological techniques continue to provide the basis for the diagnosis of both acquired and congenital toxoplasmosis [34]. Examining *T. gondii* antibodies (IgG, IgM, and IgA) and IgG avidity test results often enables doctors to determine a patient’s immunologic condition and to identify seroconversion, although toxoplasmosis is difficult to diagnose serologically, and the long-term persistence of particular IgM might make it difficult to interpret test results. The IgG avidity test can distinguish between infections that have occurred in the past and those that have just recently occurred. A low avidity test result often suggests a recent infection, whereas a high avidity test result rules out a recent infection within the previous four months [11].

## 4. Pathophysiology

There are three primary clonal lineages of *T. gondii*, strains I, II, and III. Genetic investigations have revealed that the majority of parasites found in North America, Europe, and other regions fall into one of these three closely related genotypes, which are derived from two ancestral strains [35]. Whether genotypic variations between parasites affect the infection and the degree of ocular toxoplasmosis continues to be up for debate. While type I may be more commonly observed in cases of congenital toxoplasmosis, it has been proposed that the majority of acquired ocular diseases may be caused by the type II clonal lineage of *T. gondii*. It has been demonstrated recently that type I and atypical strains may be significant factors in acquired infection [36]. In France and the US, type II strains seem to constitute the majority of cases in humans [37], which might lead to various symptoms and complications, while types I and III are detected in only 10% and 9% of patient isolates. Type I strains are the most virulent and are frequently fatal in mice infection models. Similar experimental settings only result in moderate virulence for the type II and III strains. These genetically based variations in animals are associated with certain gene loci (ROP18, ROP5, and ROP16) [38]. In Brazil, it has also been documented that type I parasite strains can cause severe postnatal ocular toxoplasmosis in patients. Vitreous body fluids from individuals who have undergone vitrectomies provide more proof that *T. gondii* type I strains cause severe fulminant retinitis. This research of Grigg et al. [39] has separated patients into two different groups; the ones infected with the type II and III strains were immunodeficient, but all patients impacted by type I strains were otherwise well and immunocompetent. These data suggest that the clinical course of this disease in immunocompetent individuals is dominated by the parasite genotype, but in immunesupressed patients, host factors play a more significant role, and any form of parasite strain can produce severe ocular pathology [39].

These three strains have been shown to proliferate throughout a broad range of geographic regions and hosts [3]. The three-strain theory may not fully capture the genetic complexity of *T. gondii*, and distinct genotypes may have a role in the diverse clinical manifestations of ocular toxoplasmosis in different regions. Additionally, the severity of the pathology has been linked to variational alleles. Understanding the genesis of ocular disease requires an understanding of the genetic basis of parasites [40].

Antibodies generated in response to *T. gondii* infection mostly target the primary surface antigen, P30. Recent research has demonstrated that only eight of the seventeen peptides that make up this protein cause mice to produce particular antibodies and that only four peptides that come from the carboxyl terminus provide protection and significantly increase survival after toxoplasmosis [41]. A few studies have emphasized how crucial it is to identify the particular sequences that successful immune responses should be aimed against. Which P30 peptides are identified by antibodies from persons infected with *T. gondii* remains unclear to this day, and many studies have exposed contradictory results [42,43]. The study of Cardona et al. [44] implied that the humoral immune responses elicited by peptides derived from the primary surface protein of *T. gondii* in humans exhibit notable variations.

The B1 gene is a highly conserved single-copy gene found in the genome of *T. gondii*. It codes for a 35-fold repeated 529-bp sequence within the parasite’s genome. Due to its high specificity, the B1 gene is widely used as a target for molecular diagnostic assays, including PCR, to detect the presence of Toxoplasma DNA in clinical samples. PCR assays targeting the B1 gene offer high sensitivity in regard to the detection of Toxoplasma DNA in ocular samples, even in cases with low parasite load or when traditional diagnostic methods, such as microscopy or serology, yield negative results [45,46]. Several studies came to the conclusion that, when compared to other targets, such as P30 and rDNA [46] and 592-bp and rDNA [47], the B1 gene would be preferred for the detection of *T. gondii* in ocular samples.

## 5. Ocular Infection and Proliferation

The main mechanism by which *T. gondii* invades ocular tissues remains uncertain. Through the use of intercellular adhesion molecule-1-mediated adhesion in its free form [48] and intercellular adhesion molecule-1, vascular cell adhesion molecule-1, and the activated leukocyte cell adhesion molecule within dentritic cells [49], in vitro research has demonstrated the ability of *T. gondii* to migrate through cells. Though it is possible that the iBRB is the favored pathway for invasion over the eBRB, a recent study revealed that parasites were more prevalent in the inner layers of the retina [50]. This finding could be the result of parasite movement inside the retina and its resident cells; however, studies have shown divergent results on which types of cells could be the preferred site of infection. Although Toxoplasma appears to be able to penetrate multiple layers of the retina before preferentially invading glial cells, studies on mice have revealed that the parasite attacks both glial and neuronal cells equally. Therefore, the detection of parasites far from their retinal site of entrance may be explained by the identification of certain cell types as being favored hosts of Toxoplasma [50].

The primary mechanism for controlling *T. gondii* growth is the production of interferon-gamma (IFNγ). In response to toxoplasmic infection, a variety of cells, including dendritic cells, macrophages, CD8+ and CD4+ lymphocytes, granulocytes, and NK cells, are implicated in the release of IFNγ. Specifically, CD4+ Th1 cells stimulated by antigen-presenting cells through IL12 generate a strong effector immunity in CD8+ T cells, which are a major source of IFNγ, and have lethal action against cells that are infected [51,52].

Th17 cells have been found to play a significant regulatory role in maintaining the equilibrium of the ocular immunological pathogenic response. These cells are thought to have proinflammatory properties and are linked to cytokines as a response to infectious and inflammatory responses [53]. In vitro studies have shown intraocular upregulation of IL-17, the most important cytokine of the Th17 lymphocytes subpopulation, in active ocular disease [54]. In a recent study, local cells within the retina rather than invading T cells were primarily responsible for the early production of this cytokine throughout infection [55]. Given that IL-17 is a recognized trigger for proinflammatory reactions and autoimmune disorders, there could be direct therapeutic and pathogenic consequences from this. However, by blocking intracellular calcium, maintaining homeostasis, and preventing apoptosis in active uveitis, IL-17 showed potent neuroprotective qualities [56]. As a result, the precise function of IL-17A in infectious illnesses is unknown, ranging from severe inflammation and tissue damage to antipathogenic action.

The immunological response during ocular toxoplasmosis in patients is suggested by a multitude of evidence. High concentrations of Th1 and inflammatory cytokines, including IL2, IL10, IFNγ, IL6, IL17, and Monocyte Chemotactic Protein-1 were found in the aqueous humor according to retrospective surveys; however, other cytokines such as IL13 were not well expressed in these patients [54,57].

## 6. Diagnostic Methods

Toxoplasmosis can be diagnosed using various techniques, including PCR, serological methods, immunohistochemistry, and identification of specific sequences in fluids and tissues. Occasionally, toxoplasmin skin tests and identification of the parasite antigen in blood and fluids are also used [58,59]. These methods are particularly helpful when the diagnosis is unclear during ophthalmoscopy. In ocular toxoplasmosis, identifying antibodies and using PCR-based tests on aqueous humor and vitreous body samples are valuable diagnostic approaches [59].

### 6.1. PCR

PCR is a sensitive molecular method for detecting *T. gondii* DNA in blood and ocular samples, such as aqueous humor, vitreous fluid, or ocular tissue biopsy specimens. Although PCR is useful for detecting Toxoplasma DNA in blood samples from patients with ocular toxoplasmosis, its sensitivity ranges from 53.3% to 75% [60]. Real-time PCR is advantageous for early diagnosis, quantifying parasite load, and differentiating between active and latent infections [61]. Nested PCR, involving two successive rounds of amplification, is commonly used to increase sensitivity and specificity [62]. Numerous investigations highlight the sensitivity and specificity of PCR-based techniques, with sensitivity ranging from 57% to 95.7% in the ocular fluids of patients with presumed toxoplasma-related uveitis [63,64]. Studies comparing the sensitivity of vitreous humor and aqueous humor found higher sensitivity in vitreous humor (95.7% vs. 81.5%) [62]. However, due to the higher risk of complications, vitreous tap should only be used in rare or complex cases, with aqueous tap being preferred for its lower risk [65].

### 6.2. Serological Methods

ELISA detects specific antibodies (IgG, IgM, IgA) against *T. gondii* antigens in patient serum or ocular fluids, aiding in the precise diagnosis and monitoring of ocular toxoplasmosis [3]. Sensitivity and specificity estimates for intraocular antibody detection are 63% and 89%, respectively [15]. The small amount of withdrawn fluid and the severe blood–aqueous barrier breakdown pose challenges to identifying specific antibodies in intraocular samples of immunocompetent patients. IgM detection is particularly useful if retinitis appears during the first year of an acquired systemic toxoplasmosis. However, ELISA’s utility is limited by the varied rate of decline of this Ig isotype. In pregnancy, maternal IgM may indicate an acute infection that requires immediate consultation [66].

The Goldman–Witmer coefficient (GWC) is a valuable test that compares specific antibody levels in ocular fluids to those in serum, providing a quantitative measure of local antibody production [67]. Studies show higher diagnostic rates for GWC compared to PCR [68,69]. For example, one study demonstrated a 62.5% positivity rate for real-time PCR and 87.5% for GWC [67]. Western Blot detects specific antibodies against *T. gondii* in patient serum or ocular fluids, with combined diagnostic methods (GWC, WB, PCR) significantly increasing sensitivity, reaching up to 97% when all three are used together [70,71,72].

### 6.3. Other Methods

Immunohistochemistry (IHC) using stained biopsies offers high specificity and quantitative analysis for diagnosing ocular toxoplasmosisv [40]. Serum levels of CXCL8 and CD25 expression can also be used for follow-up and in distinguishing acquired from congenital toxoplasmosis [73]. In vitro cultures, although not commonly used due to the difficulty in isolating *T. gondii* from ocular tissues and the slow growth rate of the parasite, are valuable in understanding the mechanisms and pathological processes of ocular toxoplasmosis [74].

Previous research on suspected human ocular toxoplasmosis in Romania reported several different laboratory tests which guided the diagnosis and management of the disease. Table 2 highlights the described investigations used by Romanian researchers.

## 7. Conclusions

Ocular disease remains one of the most important complications of *T. gondii* infection. Many patients have certainly benefited from earlier management, especially in the context of immunosuppression, as breakthroughs in molecular biology have made it easier to recognize atypical manifestations of the disease. However, important questions regarding epidemiology, the effect of parasite strains, as well as the function of the immune responses, remain unanswered despite significant advancements in the management of ocular toxoplasmosis.

There is limited information regarding human ocular toxoplasmosis in Romania. Most papers were published in local journals and were not available to researchers from other countries. Further studies on ocular toxoplasmosis in Romania could enhance our understanding of its etiology, diagnosis, and epidemiology, significantly influencing how this complex clinical entity is managed. Continued research efforts aimed at elucidating the immunopathogenesis of ocular toxoplasmosis and refining diagnostic modalities are imperative for improving clinical outcomes and reducing the global impact of this sight-threatening disease.

## Figures and Tables

**Figure 1 microorganisms-12-01541-f001:**
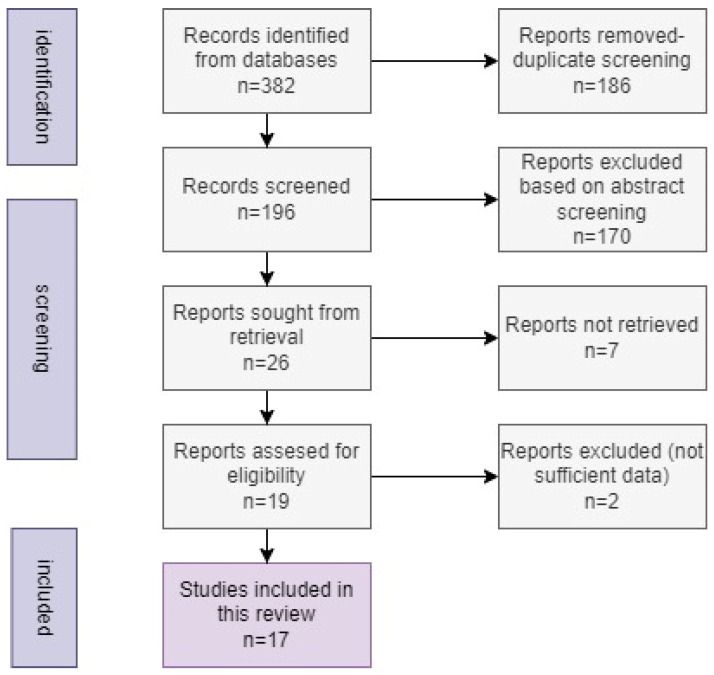
PRISMA diagram-the methodology used for the review.

**Table 1 microorganisms-12-01541-t001:** List of research articles on the prevalence of ocular toxoplasmosis in Romania.

Study	Type of Patients	Number of Patients	Number of Positive Patients	Prevalence
Proca-Cioban et al., 1981 [32]	Children with ocular Symptoms.	338	36	10.6%
Pop et al., 1992 [31]	Hospitalized patients with various eye diseases.	696	123	17.6%
Lazar et al., 2002 [25]	Hospitalized patients with retinouveal disease.	53	45	84.9%
Dogan et al., 2004 [24]	Children diagnosed with uveitis.	96	21	21.9%
Teodorescu et al., 2008 [21]	Hospitalized adult patients with ocular symptoms.	135	90	66.7%
Cobaschi et al., 2023 [33]	Hospitalized HIV patients.	1692	38	2.24%

**Table 2 microorganisms-12-01541-t002:** Group studies and case studies conducted in Romania and the investigations that were used. Abbreviations: ELISA—enzyme-linked immunosorbent; LFA—enzyme-linked immunofluorescence assay; ECLIA—electro-chemiluminescence-based immunoassay.

Study	Investigation(s)
Proca-Cioban et al., 1981 [32]	IFAT antibodies
Pop et al., 1992 [31]	ELISA IgG, IgM
Panaitescu et al., 1995 [75]	IFAT antibodies
Crucerescu et al., 1998 [76]	IFAT IgG, IgM, ISAGA IgA
Mihu et al., 1999 [29]	ELISA IgG, IgM
Stanila et al., 2003 [28]	ELISA IgG, IgM
Costache et al., 2004 [77]	ELISA IgG, IgM
Dogan et al., 2004 [24]	ELISA IgG, IgM
Silosi et al., 2006 [23]	ELISA IgM
Cretu et al., 2007 [78]	ELISA IgG, IgM
Teodorescu et al., 2008 [21]	ELISA IgG, IgM
Barca et al., 2021 [27]	LFA and ECLIA IgG, IgM
Singer et al., 2022 [26]	ELISA IgG, IgM
Cobaschi et al., 2023 [33]	ELISA IgG, IgM, Western Blot

## Data Availability

The data presented in this study are available on request from the corresponding author.

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
