# Peer review of "Human Ocular Toxoplasmosis in Romania: History, Epidemiology, and Public Health: A Narrative Review"

_microorganisms, 2024, doi:10.3390/microorganisms12081541_

Round 1

Reviewer 1 Report

Comments and Suggestions for Authors

I had the opportunity to review the paper titled ‘Ocular toxoplasmosis in Romania: history, epidemiology and

public health-a narrative review’ and I have the following comments and concerns for authors.

1/ The scope of the study seems very broad, but the information presented in the paper is very narrow and it is more of an information based on general literature. For example, explaining too much about diagnostic methods for the parasitic agent is irrelevant to the hypothesis of the study.

2/ I afraid authors have discovered all studies and clinical practices of Toxoplasmosis in Romania including its public and veterinary health importance.

3/ The scholarly contribution of this review to this field is not properly addressed.

4/ Author names should be followed by year of publication in all the tables.

5/ in the study area, any vaccination trials in humans and animals are supposed to be addressed.

6/ Overall, authors conclude that there is lack of information about the disease in the area and with such scenario, a narrative review would be an incomplete story. However, there is still a room for improvement by modifying the scope and adding an expert review and scientific justifications to the existed data.

Author Response

Dear Reviewer,

Thank you for your valuable feedback and insightful comments. We appreciate the opportunity to address your concerns and provide further clarification regarding our study.

1/ The scope of the study seems very broad, but the information presented in the paper is very narrow and it is more of an information based on general literature. For example, explaining too much about diagnostic methods for the parasitic agent is irrelevant to the hypothesis of the study.

Answer: Thank you for your comment. As suggested, we have shortened the section on diagnostic methods for human ocular toxoplasmosis.Please see the main manuscript.

2/ I afraid authors have discovered all studies and clinical practices of Toxoplasmosis in Romania including its public and veterinary health importance.

Answer: The primary aim of our study was exclusively focused on the prevalence and epidemiology of human ocular toxoplasmosis in Romania. This review specifically targeted human ocular toxoplasmosis, as this condition is notably understudied and underdiagnosed in Romania. While we recognize the broader implications of toxoplasmosis in public and veterinary health, our intention was to concentrate solely on the human ocular aspect of this zoonosis to highlight its significance.

3/ The scholarly contribution of this review to this field is not properly addressed.

Answer: By narrowing the scope to human ocular toxoplasmosis, we assessed the current situation regarding ocular toxoplasmosis in Romania and emphasized the need for improved diagnostic methods, increased research efforts, and comprehensive education tailored to this specific manifestation of the disease. We believe this targeted approach will better serve on the local level to raise awareness among healthcare professionals and researchers, encouraging further studies and public health initiatives whereas on the global level, these data may represent another piece in the jigsaw puzzle of the epidemiology of human toxoplasmosis.

4/ Author names should be followed by year of publication in all the tables.

Answer: Thank you for your suggestion. We reorganized the tables and added all of the years of publication throughout the manuscript.

5/ in the study area, any vaccination trials in humans and animals are supposed to be addressed.

Answer: We acknowledge the importance of vaccination trials in humans and animals for toxoplasmosis. However,  the aim of this work was to review  the prevalence and epidemiology of human ocular toxoplasmosis in Romania, highlighting gaps in research and diagnosis-specific to this topic.

6/ Overall, authors conclude that there is lack of information about the disease in the area and with such scenario, a narrative review would be an incomplete story. However, there is still a room for improvement by modifying the scope and adding an expert review and scientific justifications to the existed data.

Answer: As suggested, we rephrased the scope of this paper, inserted the word human in the title and rephrased the conclusions, please see the manuscript.

Reviewer 2 Report

Comments and Suggestions for Authors

• In the title, instead of the dash, we suggest putting a colon;

• On line 32, check the space between the words review and aims;

• In line 39, do not use the pronoun “we” and replace it with a pronoun with an infinitive meaning, like the authors.

• The sentence in lines 53 to 55 can be better translated into English.

• In line 60, we suggest changing the word “shed” to the word “eliminated”, giving a more formal tone to the text.

• In line 63 it is suggested to replace the phrase “As far as we are aware,” with “As far as it is known”, removing the term “we”.

• In lines 68, 74, 76, 177, 178, 180, 185, 244, 265 and 394 place the number of bibliographic references, corresponding to the cited author, following the journal's rules.

• On line 72, after “...test2-LAT2)”, insert a comma.

• On line 92, remove the point after “6%”.

• In line 129 replace the words “In this study we review”, with “Therefore, the present study aimed to review…” removing the word “we”.

• On line 130, remove a space after the word “Romania”.

• In line 132, replace the word “our” with the word “this”. In the same vein, replace the words “we hope” with “it is hoped”, removing the word “we”.

• In lines 136 to 138, replace the phrase “We followed The Preferred Reporting Items for Systematic Reviews and Meta-Analyses (PRISMA) standards during the selection of the studies that were included in this research [18]” with “The Preferred Reporting Items for Systematic Reviews and Meta-Analyses (PRISMA) standards were followed during the selection of studies that were included in this research [18]”, removing the term “we”.

• In line 144, replace the phrase “we also included case reports” with “case reports were also included”.

• On line 145, remove a space before the word “we” and replace it with the phrase “Publications with too little relevant information,...”, removing the word “we”.

• On line 146, remove a space after the word “English. “, and perform the following subsequent adjustment: “Most relevant publications were then chosen,”.

• On line 151, make the following adjustment: “Subsequently, a second search was carried out using the term MeSH…”.

• On line 158, remove a space before the word “acquired”.

• In figure one, place the caption closest to the image.

• On line 171, indicate the number corresponding to the author.

• On line 185, remove a space after “[26,27].”

• On line 189, correct “.28]”, to “symptoms [28].”

• On line 195, remove a space after “Romania was”.

• In line 197, correct “we did not include”, to “were not included”

• Table 1 can be reorganized in terms of line and paragraph spacing, offering a better and leaner look, with better use of space. It is suggested to reduce the font size (8), for example. In these cases, the journal can organize the template in a better way in terms of layout. Observing articles with tables that have already been published in the last edition of the journal, it was found that the title of the table is located above it and centered. A model is suggested below, using part of the data from the original table. The same applies to Table 2.

• On line 246, remove a space after the word “type I”.

• On line 266, remove a space after the word “T.”.

• On line 281, remove a space after the word “molecule-1”.

• On line 282, remove a space after the word “activated” and after the word “molecule”.

• On line 298, insert a comma after the word “IFNÆ´”

• On line 343, remove a space after the word “disease”.

• In line 349, after “81.5%)”, insert a comma.

• In line 352, the word “lesser” is misspelled (lesserr). To correct.

Comments on the Quality of English Language

The article is excellently written in English, requiring occasional adjustments that do not interfere with its content. Furthermore, these adjustments were indicated in the comments to the authors in a very clear and specific way.

Author Response

Dear Reviewer,

Thank you for your valuable feedback and for pointing out the typos in our manuscript. We have carefully reviewed and revised the entire document to address all the errors and typos identified. We believe these revisions have improved the clarity and readability of the manuscript. We appreciate your thorough review and hope the revised version meets your expectations.

Thank you for your consideration.

Sincerely,

The authors